# MULTIMODAL RETRIEVAL-AUGMENTED GENERATION QUESTION-ANSWERING SYSTEM

## ABSTRACT

Retrieval-Augmented Generation (RAG) combines the richness of external knowledge bases with the generative capabilities of large language models (LLMs) to provide users with more accurate and real-time responses. However, in the era of information explosion, the way information is presented is increasingly becoming multimodal. Users are no longer satisfied with the information provided by traditional text-based knowledge bases, making the construction of an efficient and accurate multimodal RAG question-answering system of significant theoretical and practical importance. To address these issues, this paper proposes an innovative RAG question-answering system: this approach pre-designs a rich dataset containing images, text, and question-answer pairs from external knowledge bases for subsequent model training, effectively improving the training quality of the model; it builds a cross-modal retrieval model from text to images, ensuring precise matching between document content and corresponding images, significantly reducing the complexity and processing time of locating relevant images within long texts. Furthermore, the retrieval model and the multimodal question-answering model are integrated to construct an efficient and accurate RAG question-answering system. Experimental results show that this system not only effectively simplifies the document formatting process and improves text-to-image retrieval accuracy but also exhibits comprehensive performance in handling multimodal data.

## 1 INTRODUCTION

Large Language Models (LLMs) have demonstrated remarkable success in performing question-answering tasks (Brown et al., 2020). By training on vast datasets, these models leverage their extensive parameterized memory to generate query responses that meet user requirements (Kojima et al., 2023). However, no training dataset can fully encompass all domains or address critical details, particularly in the current era of rapid data growth, where data iteration occurs at an exponential rate and information is increasingly presented in multimodal formats. As a result, when LLMs are required to respond to the latest information or handle knowledge-intensive questions with ambiguous factual grounding (Petroni et al., 2021), they may produce responses that are inconsistent with reality, sometimes even generating answers based on hallucinated knowledge (Huang et al., 2023).

Retrieval-Augmented Generation (RAG) (Lewis et al., 2021) has gradually become a mainstream solution in the industry to address these issues. By integrating information retrieved from external databases into the model's context (Gao et al., 2023), this approach effectively reduces factual errors in LLMs when tackling knowledge-intensive tasks. It not only enables efficient access to external, rich knowledge bases but also helps LLMs incorporate knowledge in a timely and accurate manner before responding.

Of course, this method has its limitations. The current retrieval modules often fail to achieve the desired precision for specific tasks. For instance, when answering complex questions, this method frequently requires retrieving relevant information from multiple documents to ensure that the context contains the necessary information for the response (Petroni et al., 2021). This practice increases the input length for LLMs, introducing additional delays in encoding lengthy retrieval documents and posing complex inference challenges (Ding et al., 2023). Furthermore, external databases often contain a wealth of diverse information, such as images, tables, and complex charts. The intri-

cate information embedded in charts is often difficult to convey effectively through simple textual descriptions(Kembhavi et al., 2016), and the brevity of annotations further complicates understanding. This makes it challenging for traditional text-based retrieval methods to provide precise answers when handling and interpreting multimodal data, thereby affecting the overall performance of question-answering systems. Therefore, enhancing the retrieval and generation capabilities to handle multimodal information is crucial to improving the accuracy of question-answering systems(Li et al., 2019).

In summary, to address the aforementioned issues, this paper proposes an innovative multimodal retrieval-augmented generation question-answering system. The system integrates both a text-image retrieval module and a visual multimodal model, aiming to overcome the limitations of traditional frameworks in processing chart information and to enhance the accuracy of generated answers by pre-parsing documents. The main contributions of this paper are threefold:

- Construction of a high-quality dataset: This paper designs a high quality dataset $(IMG, MD\_test, QA)$containing images, text, and question-answer pairs in advance, suitable for training multimodal models, thereby improving the system's performance in multimodal scenarios.

- Development of a text-image retrieval model: By training a retrieval model that matches text with visual information, the system achieves precise alignment between textual content and images, effectively reducing the complexity and processing time involved in locating relevant images in long texts.

- Integration of a multimodal question-answering system: This system combines cross-modal retrieval models with multimodal question-answering models to build a visual document-based multimodal RAG system. The system aims to simplify traditional document processing workflows, avoiding complex preprocessing steps (such as document parsing, OCR layout analysis, and text chunking) that may introduce errors and lead to inaccurate information transmission. It significantly simplifies the RAG document processing flow and directly answers user queries based on image content.

## 2 RELATED WORK

**Retrieval-Augmented Generation** In text-image retrieval tasks, models must align visual and textual information for effective cross-modal matching(Lu et al., 2019). Recent advancements have improved feature alignment, retrieval efficiency, and adaptability in multilingual contexts (e.g., Chinese). Recent work (Chen et al., 2023; Tan & Bansal, 2019) proposed combining global and local alignment, mapping image regions to text fragments for precise cross-modal matching, addressing the limitations of relying solely on global alignment. Retrieval efficiency remains a challenge with large datasets. Another study (Miech et al., 2021) optimized this using hash encoding and model compression, which reduces storage and computational costs but may slightly affect accuracy. Fast-slow combination strategies further balance efficiency and accuracy. In Chinese contexts, retrieval models face challenges in aligning linguistic and visual features. Recent research (Huang et al., 2020; Yu et al., 2020) enhanced performance using cross-modal pre-training and contrastive learning-based alignment methods. This paper adopts the ColPali model (Faysse et al., 2024), which generates high-quality contextual embeddings and employs post-interaction matching to enhance retrieval speed and performance, supporting end-to-end training.

**Image-Text Retrieval** In image-text retrieval, models must align visual and textual information for effective cross-modal matching. Recent work (Chen et al., 2023) proposed combining global and local alignment to improve fine-grained semantic matching, mapping image regions to text fragments for greater precision. Retrieval efficiency is a challenge with large data volumes. Another study (Miech et al., 2021) addressed this with hash encoding and model compression, reducing storage costs but potentially affecting accuracy. "Fast-slow combination" strategies also improve the balance between efficiency and accuracy. In Chinese contexts, models often struggle with text-image alignment due to linguistic and visual feature complexities. Research (Huang et al., 2020) improved performance using cross-modal pre-training and contrastive learning-based alignment. This paper uses the ColPali model (Faysse et al., 2024), which generates high-quality embeddings from document images. It applies a post-interaction matching mechanism, enhancing speed and retrieval performance, while supporting end-to-end training.

# 3 MULTIMODAL RETRIEVAL-AUGMENTED GENERATION QUESTION-ANSWERING SYSTEM

**Problem Description**: In knowledge-intensive tasks, each entry can be represented as (Q, A, D), where Q is a question or statement requiring external knowledge to answer; A is the expected answer; and D is a set of n relevant documents retrieved from an external database. In practical document retrieval, aside from textual information, there are often complex charts (e.g., line graphs, flowcharts) and even realistic images that are relevant to the question. These visual elements can effectively assist in generating the expected answer A(Kafle & Kanan, 2017). The goal of the multimodal retrieval-augmented generation question-answering system is to generate high-quality answers by using the top N most relevant images retrieved, combined with a pre-generated answer(Zhang et al., 2020).

## 3.1 OVERVIEW

This paper presents an innovative multimodal retrieval-augmented generation question-answering system designed to enhance performance in handling complex documents and tasks while maintaining processing speed. Unlike traditional methods that rely solely on text, this system integrates cross-modal retrieval and multimodal question-answering. In response to user queries, it retrieves the top K relevant images and uses a multimodal model to analyze them for direct question answering, enabling efficient text-image retrieval and intelligent question-answering.

As shown in Figure 1, the process begins by converting the document content into Markdown-formatted text along with corresponding images, and constructing a rich dataset containing these elements. By training a text-image retrieval model, optimized with contrastive learning and a cross-entropy loss function, the system ensures that the retrieval model can accurately locate relevant images from the document content based on the question. The retrieved images are then passed to the multimodal question-answering model, which fully understands the user's query and combines the information from the retrieved images to generate effective and accurate answers. In the system constructed in this paper, the text-to-image retrieval model and the multimodal question-answering model work in conjunction to achieve efficient document processing and answer generation.

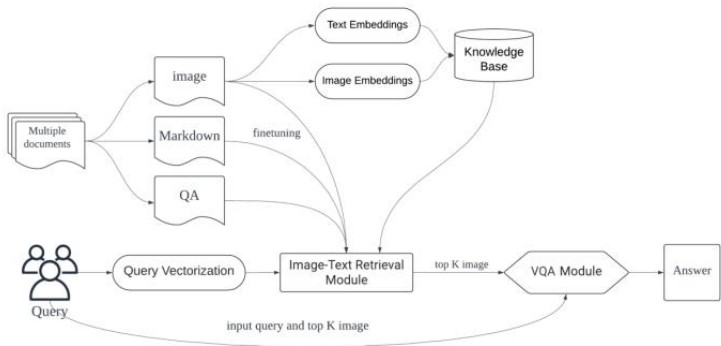

Figure 1: Flowchart of the Multimodal RAG System

## 3.2 DATASET CONSTRUCTION

Building a high-quality multimodal retrieval-augmented question-answering system depends on a diverse dataset that significantly affects model training and the knowledge base's quality(Baltrusaitis et al., 2019). Each document page from the external knowledge base is converted into Markdown format (including text, tables, and charts)(Tapaswi et al., 2016), and GPT-4o is employed to extract relevant question-answer pairs (standard text, chart, and image-based Q&A)(Raffel et al., 2020). This pre-processed and structured information allows the retrieval module to efficiently capture relevant data. The multimodal model then evaluates this content to complete the Q&A tasks. Dataset preparation involves document collection, preprocessing, and Q&A pair generation(Kwiatkowski et al., 2019).

### 3.2.1 DOCUMENT COLLECTION

The core of RAG technology is to help large language models connect with external knowledge bases, enabling them to provide rich answers for specific domains. A large number of PDFs and Word documents are collected from multiple sources to ensure that the knowledge base covers a wide range of topics and content in the field. These documents include, but are not limited to, professional papers, industry reports, white papers, and are sourced from public databases, online resources, etc.

### 3.2.2 DOCUMENT PREPROCESSING AND CONVERSION

To construct an efficient and accurate training set, information must be extracted from a large volume of PDF and Word documents, and each page of the document must undergo the following processes:

- **Markdown Format Text Conversion**: This study converts the text content of each page into Markdown format (MD_test) using layout analysis-OCR tools. Markdown is lightweight(He et al., 2020), easy to read, and easy to process, and is widely used in document writing and data processing. After the initial conversion, GPT-4o is used to format and correct the Markdown text, ensuring clear and standardized text structures that enhance the efficiency and accuracy of subsequent text processing and analysis.

- **Image Generation**:To ensure high-quality image generation for multimodal retrieval, PDF pages are first converted into high-resolution images(Deng et al., 2009). These images are optimized by adjusting resolution and cropping unnecessary edges. Data augmentation techniques, such as blurring and brightness adjustments, are applied to improve the model's generalization(Simonyan & Zisserman, 2015). Additionally, scanned document images and diverse text-image datasets are introduced to enhance performance across various scenarios(He et al., 2016). Finally, the optimized images are matched with corresponding Markdown text for efficient multimodal processing, ensuring clarity and data diversity.

- **Question-Answer Pair Generation**: This paper uses a high-quality, large-scale pre-trained model (such as GPT-4o)(Devlin et al., 2019) to generate question-answer pairs $(QA = [(q_1, a_1), \ldots, (q_n, a_n)])$ that are highly relevant to the document content from the generated Markdown text. This process covers not only textual information but also charts, flowcharts, and other types of information, ensuring that the Q&A pairs comprehensively reflect the document's overall content and details. The generated Q&A pairs have the following advantages: First, the question-answer pairs generated by the large model are more standardized and consistent in format and content expression, helping the subsequent models more effectively capture key information in the document(Brown et al., 2020). Second, the generated Q&A pairs are well-matched with the extracted text, images, and charts, ensuring the overall consistency and coherence of the dataset(Radford et al., 2019). This provides rich and high-quality data $(IMG, MD\_test, QA)$ support for subsequent model training.

### 3.3 TEXT-IMAGE RETRIEVAL MODEL

This paper uses a specialized text-image retrieval model based on the ColPali architecture, fine-tuned with a custom dataset for enhanced performance in document retrieval tasks. The model focuses on efficiently retrieving images from text, supporting the visual question-answering model and improving answer accuracy by capturing contextual information.Unlike complex semantic tasks, the model emphasizes generating high-quality contextual embeddings from document images. The ColPali model is fine-tuned to optimize the relationship between text tokens and image patches, improving precision through a late-stage interaction matching mechanism.

This approach simplifies traditional document retrieval by bypassing OCR and layout analysis, achieving faster indexing by directly processing document images. The fine-tuned ColPali model excels in retrieving visually rich content, demonstrating strong performance across multiple domains and languages.

## 3.4 CONSTRUCTION OF THE RAG QUESTION-ANSWERING SYSTEM

This section provides a detailed explanation of how to build an efficient multimodal Retrieval-Augmented Generation (RAG) question-answering system by integrating dataset construction, text-image retrieval models, and multimodal question-answering models. The system aims to leverage the advantages of multimodal information processing to provide users with accurate and comprehensive answers, demonstrating exceptional performance in handling complex document scenarios.

### 3.4.1 SYSTEM ARCHITECTURE

The entire RAG question-answering system architecture consists of two main modules: the text-image retrieval module and the multimodal question-answering module. These modules work collaboratively to cover the complete process from data input to answer output.

- **Text-Image Retrieval Module**: This module, based on the ColPali framework, builds a retrieval model to efficiently match and retrieve images from text. By comparing the user's query with the text and images in the documents, the retrieval module quickly locates relevant pages and provides them to the multimodal question-answering module for further processing.

- **Multimodal Question-Answering Module**: Based on the GPT-4o model, this module generates high-quality answers by integrating visual and linguistic features from the top NNN most relevant images retrieved. The introduction of the multimodal question-answering model greatly enhances the system's ability to handle complex queries.

### 3.4.2 SYSTEM WORKFLOW

In the actual operation of the multimodal RAG question-answering system, the system first parses the user's input query and generates embedding vectors suitable for retrieval and question-answering tasks. This step leverages a pre-trained language model to generate high-dimensional embedding representations, ensuring that the query content is accurately captured to support the subsequent retrieval process. The system calculates the similarity between the query vector and image vectors to retrieve relevant documents from the external knowledge base, using the retrieval model to extract relevant images or pages associated with the query.

During this process, the retrieval module not only integrates the semantic information from the text but also incorporates visual information, ensuring that the results reflect the document content comprehensively. The retrieved images are then passed to the GPT-4o multimodal model for processing. This model can deeply analyze the images and, in conjunction with the textual information contained within the images, generate answers that are highly consistent with the query context. By precisely aligning visual and textual features, the GPT-4o model ensures that the answers fully reflect the critical information in the images while maintaining coherence and accuracy. Finally, the system optimizes the generated answers, including format adjustments, filtering of redundant information, and enhancing linguistic coherence to ensure that the user receives the highest-quality answer output. The detailed algorithm workflow is shown in Table 1.

Table 1: *Algorithm of Multimodal QA System*

**1. Input Query:**
- Convert the user *query* into $\{q_{embed}\}$.

**2. Retrieve Relevant Images:**
- Use the *pre-tuned* model R to retrieve the top N images $\{img_1, ..., img_N\}$ by comparing $\{q_{embed}\}$ with $\{text_{embed}, img_{embed}\}$.

**3. Pass Results to VQA Model M:**
- Input $\{img_1, ..., img_N\}$ and *query* into model M.

**4. Generate Answer:**
- Use M to generate the answer based on *query* and $\{img_1, ..., img_N\}$.

**5. Output:**
- Return the generated *answer* along with images $\{img_1, ..., img_N\}$.

### 3.4.3 KEY TECHNICAL DETAILS

The system's key technical features include cross-modal alignment, optimization of retrieval and generation efficiency, and ensuring robustness in question-answer generation. By progressively freezing and adjusting pre-trained layers, the system aligns text and image information, enhancing the fusion of features for more accurate answers. To handle long texts and complex images, late-stage interaction matching and visual-language embedding optimization are employed, reducing delays and improving performance. Multiple rounds of model optimization ensure robustness across domains and languages, while adaptive formatting for charts and flowcharts further improves accuracy and consistency.

## 4 EXPERIMENTAL EVALUATION AND RESULTS

This chapter provides an experimental evaluation of the proposed multimodal RAG question-answering system across four key dimensions: retrieval accuracy, generation quality, response speed, and multimodal consistency. Using a multimodal document dataset, metrics such as Precision@K(Karpukhin et al., 2020), F1 Score, BLEU(Papineni et al., 2002), and ROUGE(Lin, 2004) are employed to objectively compare the system's performance against classical models. The results demonstrate that the proposed system significantly surpasses baseline models in processing multimodal information and generating high-quality content, while maintaining fast response times.

### 4.1 EXPERIMENTAL SETUP

#### 4.1.1 DATASET DESCRIPTION

The experimental dataset comprises multimodal documents containing text, images, and question-answer pairs, sourced from diverse domains such as finance, law, and healthcare. These documents, obtained from public databases, research papers, and industry reports, are described in detail in Chapter 3. The dataset includes over 50,000 pages in formats like PDF and Word, which have been converted into structured Markdown text with high-resolution images. To support the system's question-answering functionality, each page is paired with relevant question-answer sets that reflect the complexity of the content.

To ensure the model's generalization ability, the dataset is split into training, validation, and test sets, with a ratio of 80%, 10%, and 10%, respectively. This division helps to provide more representative performance across different datasets, prevents overfitting(Goodfellow et al., 2016), and ensures the model's adaptability across different domains and scenarios(Kohavi, 1995).

#### 4.1.2 EXPERIMENTAL ENVIRONMENT

The experimental environment configuration is shown in Table 2.

Table 2: Experimental Environment Configuration

| Environment Name | Configuration |
|---|---|
| Operating System | Ubuntu 18.04.6 LTS |
| CPU | Intel(R) Xeon(R) Platinum 8369B CPU @ 2.90GHz * 128 |
| Memory | 1.0TB |
| GPU | NVIDIA A100-SXM4-80GB * 8 |
| Programming Language | Python 3.10 |

### 4.1.3 EVALUATION METRICS

The following metrics were used to evaluate the system's performance:

- **Retrieval Precision (Precision@K)**:Precision@K measures the proportion of correct images in the top K results, reflecting the retrieval model's accuracy in identifying relevant images within complex documents.The calculation method is shown in Equation (1):

$$\text{Precision} = \frac{tp}{tp + fp} \tag{1}$$

- **Generation Quality**:This metric assesses the accuracy and coherence of generated answers compared to reference answers using metrics like F1 Score, BLEU, and ROUGE, reflecting the model's ability to generate high-quality answers from multimodal data. The calculation methods are shown in Equations (2), (3), (4), and (5):

$$F1 = \frac{2 \times (\text{Precision} \times \text{Recall})}{\text{Precision} + \text{Recall}}, \quad \text{Recall} = \frac{tp}{tp + fn} \tag{2}$$

$$ROUGE - N = \frac{\sum_{n\text{-gram}\in Reference} \text{Match}(n\text{-gram})}{\sum_{n\text{-gram}\in Reference} \text{Total}(n\text{-gram})} \tag{3}$$

$$ROUGE - L = \frac{LCS(\text{Candidate}, \text{Reference})}{\text{Length of Reference}} \tag{4}$$

$$BLEU = \text{BP} \times \exp\left(\sum_{n=1}^{N} w_n \log p_n\right) \tag{5}$$

- **Latency**:Latency measures the average time taken from receiving a query to generating a response, assessing the system's efficiency in processing multimodal data and ensuring real-time performance with accuracy.

- **Multimodal Consistency Score (MCS)**:This metric assesses the consistency between generated answers and input text and images. MCS reflects the model's ability to align and integrate cross-modal information, indicating the coherence and reasonableness of the generated results. The calculation method is shown in Equation (6), where the text modality embedding is $E_T$ and the image modality embedding is $E_I$:

$$\text{Cosine Similarity} = \frac{E_T \cdot E_I}{\|E_T\| \times \|E_I\|} \tag{6}$$

## 4.2 EXPERIMENTAL DESIGN

The experimental process begins with data preprocessing, where documents are converted into Markdown text and high-resolution images, followed by generating question-answer pairs for structured training. The ColPali-based text-image retrieval model is trained using contrastive learning and a cross-entropy loss function to enhance multimodal tasks. Precision@K is used to evaluate retrieval accuracy, while generation quality and multimodal consistency score (MCS) assess answer accuracy and alignment. The average response time measures system efficiency, ensuring a balanced evaluation of preprocessing, retrieval, generation, and real-world performance.

### 4.2.1 BASELINE MODELS

To verify the effectiveness of the system proposed in this paper, the following three baseline models were selected for comparison:

- **Text-Only RAG (DPR + T5)**:We selected Dense Passage Retrieval (DPR) combined with T5 as a baseline model(Karpukhin et al., 2020). DPR retrieves documents using dense embeddings based on query similarity. T5 generates relevant answers from these documents and excels in open-domain question answering tasks(Raffel et al., 2020).

- **Haystack 2.0-based RAG System (EasyOCR + FAISS + T5)**: This baseline model integrates EasyOCR for text extraction, FAISS for vector retrieval, and T5 for answer generation within the Haystack 2.0 framework(Faisal et al., 2020), efficiently processing multimodal documents for question-answering tasks.

- **Chinese-CLIP-based RAG System (Chinese-CLIP-RAG)**:This baseline model employs Chinese-CLIP for precise text-image alignment and retrieval(Radford et al., 2019), coupled with GPT-4o for answer generation. While effective in Chinese multimodal contexts requiring accurate text-image retrieval, its tightly coupled retrieval and generation modules may introduce efficiency bottlenecks in complex scenarios.

### 4.3 COMPARATIVE EXPERIMENTAL RESULTS AND ANALYSIS

This section presents a detailed comparison of the proposed system with baseline models in terms of retrieval accuracy, generation quality, and system efficiency. By comparing the proposed multimodal RAG system with various baseline models, we assess the improvements in processing multimodal documents and the overall system performance.

### 4.3.1 PERFORMANCE ANALYSIS OF THE RETRIEVAL MODULE

In the performance analysis of the retrieval module, the proposed multimodal RAG system demonstrated significant advantages in three key metrics: Precision@1, Precision@3, and Precision@5, achieving 82.3%, 78.6%, and 75.4%, respectively. In contrast, the Text-Only RAG system had a Precision@1 of only 60.5%, with Precision@3 and Precision@5 at 55.8% and 53.1%, respectively, showing a gradual decline in accuracy due to the lack of multimodal information processing.

The Haystack 2.0 system, which integrates OCR technology to improve its ability to handle visual information within documents, achieved a Precision@1 of 68.4%, with Precision@3 and Precision@5 at 63.2% and 61.0%, respectively. While this system performed better than the Text-Only RAG, it still fell short compared to the proposed system.

The Chinese-CLIP RAG system performed well in the text-image alignment task, achieving a Precision@1 of 80.2%, with Precision@3 and Precision@5 at 76.5% and 72.9%, respectively. Although this system achieved good results in multimodal retrieval, it still slightly underperformed compared to the proposed system.

Table 3 provides a detailed comparison of the retrieval precision results for each model, indicating that the proposed system significantly improves the ability to locate images and charts within documents by incorporating efficient cross-modal alignment strategies and a text-image retrieval model. This is especially advantageous when handling complex multimodal data. As a result, the proposed system not only enhances retrieval accuracy in multimodal documents but also reduces the false positive rate, demonstrating higher precision and robustness.

Table 3: Comparison of Retrieval Accuracy Across Models

| Model | Precision@1 | Precision@3 | Precision@5 |
|---|---|---|---|
| Text-Only RAG | 0.605 | 0.558 | 0.531 |
| Haystack2.0 RAG | 0.684 | 0.632 | 0.610 |
| ChineseCLIP RAG | 0.802 | 0.765 | 0.729 |
| Our Model | 0.823 | 0.786 | 0.754 |

### 4.3.2 COMPARISON OF QUESTION-ANSWER GENERATION QUALITY

In the task of question-answer generation, our proposed multimodal RAG system significantly outperforms the baseline models in terms of generation quality. As shown in Table 4, our model achieved an F1 Score of 72.1, a ROUGE-L score of 32.4, and a BLEU score of 7.9. In comparison, the Text-Only RAG model obtained an F1 Score of 55.2, a ROUGE-L score of 26.8, and a BLEU score of 6.1. The ChineseCLIP-RAG model achieved an F1 Score of 70.4, a ROUGE-L score of 29.2, and a BLEU score of 7.4. These results demonstrate that our model provides a notable improvement in generation quality over the baseline models.

Specifically, the Text-Only RAG model lacks the ability to comprehensively process multimodal information, leading to lower accuracy and coherence in the generated answers. Although the Haystack 2.0 RAG system integrates OCR technology, the generated text content remains somewhat incoherent. In contrast, our proposed system integrates both retrieved image and text information, significantly enhancing the accuracy of the generated answers and their relevance to the questions, particularly excelling in handling complex charts and flowcharts.

Furthermore, our system's ability to understand visual information during answer generation far exceeds that of the other baseline models. This enables it to produce answers that are not only accurate but also fully reflective of the multimodal information contained within the documents, thereby providing a more comprehensive and coherent response.

Table 4: Comparison of Generation Quality Across Models

| Model | F1 Score | ROUGE-L | ROUGE-1 | ROUGE-2 | BLEU |
|---|---|---|---|---|---|
| Text-Only RAG | 55.2 | 26.8 | 26.1 | 8.1 | 6.1 |
| Haystack 2.0 RAG | 62.7 | 27.9 | 28.3 | 9.3 | 7.2 |
| ChineseCLIP-RAG | 70.4 | 29.2 | 31.0 | 13.1 | 7.4 |
| Our Model | 72.1 | 32.4 | 31.8 | 13.8 | 7.9 |

The Multimodal Consistency Score (MCS) is a key metric for evaluating the alignment between generated answers and multimodal input, such as images, tables, and flowcharts. MCS assesses the model's ability to accurately integrate visual information into its answers.As shown in Table 5, the MCS results reveal the different models' capacities to handle multimodal information. The Text-Only RAG model scores low in MCS, as it cannot process visual content effectively, limiting the inclusion of images or tables in its answers. The Haystack 2.0 RAG model, with OCR integration, shows improvement but still struggles with complex multimodal tasks, resulting in average MCS performance.The Chinese-CLIP RAG model demonstrates better alignment and integration of text and images, improving its MCS score. However, it faces challenges in tasks requiring more complex information integration due to its tightly coupled retrieval and generation modules.

Our proposed multimodal RAG system achieves the highest MCS score, thanks to optimized text-image alignment and cross-modal fusion strategies. In tasks involving complex charts and flowcharts, our model generates more consistent and coherent answers, highlighting the importance of multimodal consistency.

Table 5: Comparison of Multimodal Consistency Scores Across Models

| Model | MCS Score |
|---|---|
| Text-Only RAG | 35.2% |
| Haystack 2.0 RAG | 57.8% |
| Chinese-CLIP RAG | 68.4% |
| Our Model | 71.9% |

### 4.3.3 SYSTEM RESPONSE SPEED ANALYSIS

Response speed (latency) is a key metric for evaluating multimodal question-answering systems, as it directly affects user experience. This experiment compares the latency of Text-Only RAG, Haystack 2.0 RAG, Chinese-CLIP RAG, and our proposed multimodal RAG system.The results indicate that Text-Only RAG has the fastest response time due to lower computational complexity, but its inability to handle images and tables reduces its accuracy in multimodal tasks. Haystack 2.0 RAG, with OCR integration, improves multimodal accuracy but suffers from slower response times due to added computational demands. Chinese-CLIP RAG offers high retrieval accuracy, but its tightly integrated retrieval and generation modules lead to longer response times.

Our proposed multimodal RAG system, by optimizing text-image alignment and reducing computational redundancy, achieves a fast response time of 1.8 seconds while maintaining high generation quality, outperforming other multimodal models.

## 5 CONCLUSION

This paper proposed a multimodal retrieval-augmented generation question-answering system that integrates a high-quality dataset, a image-text retrieval model, and a multimodal question-answering model. Experimental results demonstrate that the system excels in retrieval accuracy, generation quality, and response speed when processing complex documents, particularly in chart-intensive scenarios, greatly enhancing the user experience.

Despite these positive results, the system still has room for optimization in handling complex multimodal data, such as further improving retrieval accuracy and response speed. Future work will focus on addressing these challenges and advancing the application of multimodal question-answering systems in more real-world scenarios.

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
