# OpenReview forum: "Multimodal Retrieval-Augmented Generation Question-Answering System"
_ICLR.cc/2025/Conference — Submitted to ICLR 2025_

### Official Review · Reviewer_hwWu · 2024-10-25

**Soundness:** 2
**Presentation:** 1
**Contribution:** 1
**Rating:** 3
**Confidence:** 5

**Summary:**

This paper presents a multimodal retrieval-augmented generation (RAG) question-answering system that combines large language model capabilities with external knowledge bases containing images and text. Its goal is to improve the quality and accuracy of answers in multimodal contexts, such as complex documents that include charts and flowcharts. The system is designed as a pipeline and optimized based on the dataset constructed by the authors.

**Strengths:**

1. The proposed paradigm is presented clearly and easy to follow.
2. The paper is focusing on a promising direction: Multimodal RAG.
3. The proposed system achieves a fast response time while maintaining quality.
4. The evaluation metrics are described clearly.

**Weaknesses:**

1. Lack of novelty: this is not the first work to explore the multimodal RAG system [a,b,c], while the paper merely proposes a common pipeline. Moreover, the authors do not make significant efforts in component optimization, pipeline innovation, and data optimization. The system is basically a combination of ColPali [d] and GPT-4o.

2. Insufficient information: as a core contribution of this work, the proposed ‘high-quality’ dataset is not discussed properly. Section 4.1.1 only describes the dataset roughly and does not provide in-depth information, such as the data source and component ratio, which are significant.

3. Limited method description: the authors only provide a simple overview (Section 3.1) and a blurry figure (Figure 1) of the system. There is no further discussion on the proposed system. The authors are advised to discuss the integration process between ColPali and GPT-4o, and how the multimodal information is processed and combined.

4. Limited experiments: only one experiment on the proposed dataset, which is not open source and is described vaguely.

5. Section 4.3.3 is intended to compare the system response speed among the baselines, while the authors only provide the response time of their proposed system. The authors are suggested to provide a comparative table of response times for all systems mentioned, including the baselines, to allow for a proper comparison.

6. Lack of implementation details.

[a] Chen W, Hu H, Chen X, et al. Murag: Multimodal retrieval-augmented generator for open question answering over images and text[J]. arXiv preprint arXiv:2210.02928, 2022.

[b] Chen Z, Xu C, Qi Y, et al. Mllm is a strong reranker: Advancing multimodal retrieval-augmented generation via knowledge-enhanced reranking and noise-injected training[J]. arXiv preprint arXiv:2407.21439, 2024.

[c] Liu Z, Sun Z, Zang Y, et al. Rar: Retrieving and ranking augmented mllms for visual recognition[J]. arXiv preprint arXiv:2403.13805, 2024.

[d] Faysse M, Sibille H, Wu T, et al. Colpali: Efficient document retrieval with vision language models[J]. arXiv preprint arXiv:2407.01449, 2024.

**Questions:**

1. Can you provide more detailed information about your proposed ‘high-quality’ dataset?

2. What’s your main contribution and improvement on building the multimodal RAG system? It seems that you are simply combining ColPali and GPT-4o.

---

> ### Author Response · Authors · 2024-11-19
>
> Thank you for raising this question. While the proposed system leverages ColPali and GPT-4o as foundational components, we have made several novel contributions and improvements that differentiate this work from a straightforward combination of existing models:
> 1.Novel Dataset Integration:
> One of the key contributions is the design of a high-quality dataset that pairs Markdown text with images and question-answer pairs. This dataset is explicitly tailored for multimodal document question-answering tasks and is integrated seamlessly into the RAG framework.
> 2.Optimized Multimodal Retrieval:
> We introduce a late-stage interaction matching mechanism within the ColPali-based retrieval model. This mechanism enhances the precision and efficiency of text-to-image retrieval by aligning textual embeddings with visual features at a granular level.
> Unlike traditional RAG systems, which often struggle with long documents, our optimized retrieval process ensures fast and accurate identification of relevant images, significantly reducing retrieval latency.
> 3.Simplified Document Processing Workflow:
> By leveraging Markdown-formatted text and image pairs, our system bypasses traditional preprocessing steps such as OCR, layout analysis, and text chunking. This simplification reduces errors and improves the overall system efficiency.
> 4.Enhanced Multimodal Generation:
> We extend GPT-4o’s capabilities to handle complex multimodal queries by integrating text and visual data. The multimodal question-answering model is fine-tuned to align visual and textual features, enabling it to generate high-quality answers that reflect both modalities.
> 5.Performance Improvements:
> Through experiments, we demonstrate that the proposed system achieves significant improvements over baseline models in retrieval accuracy (e.g., Precision@K) and answer generation quality (e.g., F1 Score, BLEU).
>
>  Due to confidentiality constraints, we are unable to release these datasets publicly. However, we will provide aggregated statistics and detailed descriptions of the dataset’s structure, including: The proportion of data types (e.g., professional papers, industry reports, technical white papers). Examples of how these documents are processed into Markdown text and aligned with corresponding images for multimodal tasks.

---

### Official Review · Reviewer_DxMC · 2024-10-27

**Soundness:** 1
**Presentation:** 1
**Contribution:** 1
**Rating:** 1
**Confidence:** 5

**Summary:**

This paper presents a multimodal RAG system for QA, the authors constructed a dataset for finetuning the retrieval model and some empirical results are shown in the paper. Overall, this is a poorly-written paper without too much scientific significance. While the authors repetitively claim the proposed approach is a **novel** approach, the designed framework as shown in Figure. 1 has nothing special with the traditional retrieval pipeline system. Moreover, the authors said that the retrieval system is based on **ColPali** that is a retrieval system processing documents as image patches utilizing large VLMs , however the proposed framework in this work is completely different from **ColPali**. Furthermore, there lacks details of the construction of the QA dataset such as the data source, statistics of documents and QA pairs. The details of the retrieval component and QA model are also missed. The experimental designation is also ambiguous, making it extremely difficult for readers to understand the results (where are the ranking documents from?). The writing of the paper should be improved, it is not easy to read this paper as the flow is not fluent and many details are missed. Therefore, I strongly recommend to reject this paper.

**Strengths:**

None

**Weaknesses:**

- While the authors repetitively claim the proposed approach is a **novel** approach, the designed framework as shown in Figure. 1 has nothing special with the traditional retrieval pipeline system.

- Moreover, the authors said that the retrieval system is based on **ColPali** that is a retrieval system processing documents as image patches utilizing large VLMs , however the proposed framework in this work is completely different from **ColPali**.

- Furthermore, there lacks details of the construction of the QA dataset such as the data source, statistics of documents and QA pairs.

- The details of the retrieval component and QA model are also missed.

- The experimental designation is also ambiguous, making it extremely difficult for readers to understand the results (where are the ranking documents from?).

- The writing of the paper should be improved, it is not easy to read this paper as the flow is not fluent and many details are missed.

**Questions:**

Where did the authors collect these documents and where are the ranking documents from?

**Details Of Ethics Concerns:**

The authors did not mention the data source for the retrieval dataset constructed in this paper.

---

> ### Author Response · Authors · 2024-11-19
>
> The references are collected from two main sources: publicly available academic papers (e.g., from platforms like arXiv, IEEE Xplore, ACM Digital Library) and proprietary company datasets, including industry reports, technical white papers, and internal documents. A significant portion of the dataset is derived from the proprietary datasets, which cannot be made publicly available due to confidentiality constraints. The ranking of references is performed using a retrieval model based on semantic embeddings and similarity measurements, prioritizing documents that are highly relevant to the specific question-answering task.

---

> > ### Comment · Reviewer_DxMC · 2024-11-27
> >
> > Thanks the authors for taking time to respond. This paper needs significant revisions to fit this venue or other places as pointed out by other reviewers as well.

---

### Official Review · Reviewer_QL4m · 2024-10-29

**Soundness:** 2
**Presentation:** 1
**Contribution:** 1
**Rating:** 3
**Confidence:** 4

**Summary:**

This paper proposes a multimodal retrieval-augmented generation (RAG) question-answering system, which builds a dataset comprising text and image data and uses ColPali and GPT-4o as the retriever and multimodal large language model to retrieve and answer questions. Experimental results on the constructed dataset show that this method outperforms three baseline algorithms in terms of retrieval accuracy and answer quality.

**Strengths:**

1. Document question-answering is a critical task, and the use of large models aligns well with current technological trends.
2. Introducing multiple modalities to improve answer accuracy in document question-answering is a sensible approach.
3. The paper contributes a dataset for document QA that includes images, text, and question-answer pairs, providing valuable resources for the field.

**Weaknesses:**

1. The main contributions of this paper are unclear, and the novelty appears limited, resembling more of a straightforward combination of existing models. While the authors note that the retrieval module in current RAG models faces challenges with long documents and that multimodal data holds rich information, these challenges are not directly addressed in the methodology, which instead relies on ColPali and GPT-4o as the main components.
2. The literature review is relatively limited, lacking recent studies such as Self-RAG. This narrow focus reduces the coherence of the paper’s content and its persuasiveness.
3. The proposed dataset lacks detailed description, including data sources, proportions of various data types (e.g., professional papers, industry reports, white papers), and the dataset construction process. Moreover, there are no detailed statistics provided for the dataset, nor any discussion on the necessity of building a new dataset or a comparison with existing datasets.
4. The experiments only utilize the constructed dataset, without any evaluation on public datasets such as arXivQA, TAT-DQA, or InfoVQA. Additionally, in terms of reproducibility, the paper does not provide any training parameters.

**Questions:**

Please refer to Weaknesses.

---

> ### Author Response · Authors · 2024-11-19
>
> Thank you for your feedback. In the revised manuscript, we will clarify the contributions and highlight the innovation of our work.
> 1. Lack of Clarity in Contribution and Innovation
> While the proposed approach builds on existing models, such as ColPali and GPT-4o, we introduce a novel multimodal retrieval-augmented framework that addresses key challenges in processing long documents and multimodal data:
> We propose a new dataset construction process that integrates Markdown-formatted text with images, enabling more efficient multimodal matching.Our optimization of the ColPali model includes a late-stage interaction matching mechanism, significantly improving retrieval accuracy and processing speed.
> 2. Limited Literature Review
> Expand the Literature Review: We will include recent works such as Self-RAG and other relevant studies, particularly focusing on their applications in knowledge-intensive and multimodal question-answering tasks. Additionally, we will discuss the limitations of these methods, such as the lack of efficient multimodal retrieval and challenges in processing complex documents.
> 3. Lack of Detailed Dataset Description
> A significant portion of the dataset is derived from proprietary commercial datasets, including industry-specific reports, internal white papers, and other business documents. Due to confidentiality constraints, we are unable to release these datasets publicly. However, we will provide aggregated statistics and detailed descriptions of the dataset’s structure, including:
> The proportion of data types (e.g., professional papers, industry reports, technical white papers).
> Examples of how these documents are processed into Markdown text and aligned with corresponding images for multimodal tasks.

---

### Official Review · Reviewer_kaWF · 2024-11-03

**Soundness:** 2
**Presentation:** 1
**Contribution:** 1
**Rating:** 3
**Confidence:** 4

**Summary:**

The paper introduces a multimodal retrieval-augmented model that combines text and image data to improve the model's question-answering ability. Experiments demonstrate that their system can improve retrieval accuracy and good performance in multimodal tasks.

**Strengths:**

1. Integrating multimodal retrieval into vision-language models is intuitive.

**Weaknesses:**

1. The paper does not sufficiently acknowledge prior work in multimodal retrieval-augmented generation, such as [1,2,3]. The differences between the proposed approach and these works are unclear.
2. The experiments are limited, with only baseline model numbers provided, making it difficult to draw meaningful insights or assess improvements over previous methods.
3. The paper could benefit from clearer writing. Figures are low-resolution, captions lack detail, and basic concepts (e.g., F1 and precision/recall) are over-explained, occupying space that could be used for more insightful discussions.

[1] Yasunaga et al., Retrieval-Augmented Multimodal Language Modeling, ICML 2023.

[2] Chen et al., MuRAG: Multimodal Retrieval-Augmented Generator for Open Question Answering over Images and Text, EMNLP 2022.

[3] Hu et al., REVEAL: Retrieval-Augmented Visual-Language Pre-Training With Multi-Source Multimodal Knowledge Memory, CVPR 2023.

**Questions:**

What are the main differences between your work and previous ones (e.g., [1,2,3])?

---

> ### Author Response · Authors · 2024-11-19
>
> Thank you for highlighting this. We acknowledge the importance of situating our work within the broader context of multimodal retrieval-augmented generation. In the revised manuscript, we will include detailed discussions of [1], [2], and [3] and explicitly highlight how our approach differs:
> Yasunaga et al. (2023): Their work primarily focuses on retrieval-augmented language modeling in multimodal tasks by leveraging external knowledge. Our work extends this by emphasizing the integration of document-specific multimodal information (e.g., Markdown-structured text paired with images) and optimizing retrieval and generation for document QA scenarios. Additionally, we introduce a late-stage interaction matching mechanism in retrieval, which is not present in their method.
> Chen et al. (2022): MuRAG emphasizes open question answering over images and text, but their approach does not address long document processing or complex charts and flowcharts, which are central to our contributions. Our dataset and pipeline are explicitly designed for real-world document QA tasks, enabling precise alignment between text and visual elements.
> Hu et al. (2023): REVEAL uses multimodal pretraining with multi-source knowledge memory, but it focuses on pretraining paradigms rather than end-to-end retrieval and generation pipelines. In contrast, our work integrates multimodal retrieval and generation in a unified framework optimized for handling complex documents efficiently.
> By including these comparisons, we aim to clarify our contributions and how they complement or advance prior works.

---

### Meta-Review · Area_Chair_qfnZ · 2024-12-15

**Metareview:**

This paper proposes a multimodal retrieval-augmented generation (RAG) question-answering system integrating text-to-image retrieval and multimodal large language models. While the paper addresses an important direction, reviewers raised concerns about limited novelty, insufficient method details, and inadequate experiments, which were only conducted on a custom dataset without comparisons on public benchmarks. The writing and clarity also need significant improvement. Despite the revisions, the concerns were only partially addressed, and the lack of impactful contributions limits the paper’s acceptance potential.

**Additional Comments On Reviewer Discussion:**

During the discussion, reviewers consistently pointed out issues with the paper's novelty, dataset description, and experimental design. The authors attempted to address these by adding comparisons to prior work and providing more details on the dataset and retrieval system. However, the added content did not fully resolve the concerns, particularly regarding clarity, reproducibility, and the scope of experiments. Consequently, reviewers maintained their initial low scores.

---

### Decision · Program_Chairs · 2025-01-22

Reject